# Accomplice Neighborhood: Everyday Life Politics

Héctor Fernández Medrano 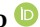

Facultad de Letras, University of the Basque Country, 01006 Vitoria-Gasteiz, Spain; hecfermed@gmail.com

**Abstract:** This paper delves into the conceptual delineation of the institution of the neighborhood as a catalyst for innovative political discourse and practice. It aims to set the basis for an upcoming reevaluation of the work of Andrés Ortiz-Osés, pioneer of Gadamerian hermeneutics in Spain, considering the neighborhood's potential: its co-implicated and co-implicative nature connects consistently with his symbolic hermeneutics of the sense. The neighborhood, a complex institution transcending conventional affiliations, underpins coexistence, mutual tolerance, and a kind of ethical dialogue. This work contributes to highlighting the neighborhood's political dimensions on its own and claims its philosophical relevance beyond its traditional understanding. The ambivalent space of vicinity promotes plural speech and serves as a vital agora, fostering dynamic, ethical coexistence and engaged citizenship, thereby enhancing the democratic landscape.

**Keywords:** neighborhood; co-implication; alterity; community; democracy; Ortiz-Osés

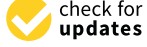



## 1. Introduction

This text is an attempt at a conceptual delineation of the institution of the neighborhood. It could play a relevant role in an innovative political reading of Basque-Aragonese author Andrés Ortiz-Osés, a symbologist and introducer of Gadamerian hermeneutics in Spain.

Ortiz-Osés studied Theology at the Pontifical University of Comillas and Philosophy at the Gregorian University in Rome. He obtained his doctorate in Hermeneutics in Innsbruck, having been a student of Gadamer, Emerich Coreth, and Franz-Karl Mayr. In contact with the Jungians of the Eranos Circle, he developed his particular philosophical approach. He was known for providing a consistent and unitary interpretation of the abundant Basque ethnographic material, following the matriarchalist framework of Bachofen. He carried out his teaching duties primarily in the areas of metaphysics and cultural hermeneutics as a professor at the University of Deusto (Bilbao). In the Spanish-speaking world, he has been placed in the philosophical vicinity of contemporaries such as Eugenio Trías, with his "philosophy of the limit", and Mauricio Beuchot, with "analogical hermeneutics".

Assuming the radical linguistic condition of the human world, its fundamental concern will be that of the sense of existence. This will lead him to pay special attention to the great symbols, where sense/meaning appears in its radical ambivalence and problematic nature. He will address this question through the category of "implication". This will restructure classic dialectic movement as "dualectic", the ambiguous logic beneath symbolic meaning. This will lead to the defense of an affective and culturalistic use of reason, involving the co-implication of the opposites.

In his symbolic hermeneutics, Ortiz-Osés typically distinguishes three major worldviews, each with its associated values: the matriarchal-naturalistic (positively communal, negatively anti-individualistic), the patriarchal-rationalistic (positively progressive, negatively abstractionist), and the fratriarchal-personalist, whose symbol is the person (which would reconcile the previous ones within itself).

His fraternal, personalistic proposal of a "fraterland" would gain sociopolitical density, moving it away from the archetypal symbolism of the family he was used to. I believe that the ontological loans implied by the metaphorical framework of the family could be

compensated by rethinking this author's ethical-political statements from the fundamental anthropological experience of neighborhood. The doctoral theses written about this author (Guerenabarrena, F., Minaya, E., etc.), and those who have worked monographically on his thought (Garagalza, L., Solares, B., etc.), have not attempted this kind of specifically political approach.

The neighborhood would serve as an interesting ontological and argumentative anchor to Ortiz-Osés's promise of a "postmodern fraternal mythology" focused on the "archetypal figure of the 'accomplice' as a co-implicationist of the Other" [1] (p. 13). If this is to be later analyzed, taking into account the archetypal status of the said figure and its ability to operate as a philosophical and political category, it is mandatory to minimally clarify what the neighborhood is.

The neighborhood as a space, a set of relationships, and a project, shapes our lives and conditions our expectations. It is governed by different criteria than filiation or parentage; it deactivates a certain logic of origin, and needs to be rethought philosophically. Even with the minimal intent of distinguishing it from categories that are confused or overlap with it (be it the community, the family, etc.). It is, on the other hand, both the stage and the matter of politics understood in everyday terms and far from orthodox partisanship. Let these pages serve as the starting point for this reflection.

This work tries to innovate in two ways. In the more regional context of the study of Ortiz-Osés' symbolic hermeneutics of sense, no research has assumed a primarily political take; much less the attempt to project it onto the intermediate social institution of the neighborhood. In a broader context, the philosophical approach to the neighborhood as something more than a condition of possibility for community and politics is beginning to fill a space for a relatively unexplored philosophical reflection.

The central category of this work, neighborhood, appears to say 'more' in Spanish, the original language of this text, than it does in English. Or at least, it does so more prominently as both 'vecindario' (neighborhood) and 'vecindad' (vicinity) share the same Latin root. This is why "Neighborhood" will be consistently used, even if the idea of closeness or vicinity is only subtly present in it as compared to the Spanish concept. All quotations have been translated from their source languages.

## 2. The Ambivalence of Neighborhood

"(. . .) men cannot disperse to infinity across the globe, whose surface is limited, and therefore must mutually tolerate each other's presence, since originally, no one has a better right than another to be in a particular place on the planet". [2] (p. 36)

Neighborhood is a fundamental circumstance, far from our choice, that radically binds our lives. It is an ambivalent intermediate social institution, although its relationships are fundamentally defined by annoyances, discomforts, and disturbances. What intrusions from alien activities should be endured and which should be eliminated are often determined in this framework, preferably without effort or coercion. A sort of mutual tolerance ends up imposing itself due to mutual necessity [3] (pp. 23,168). The essential task of coexisting is outlined from this condition and, simultaneously, the ceaseless search for that same condition. Neighborhood promotes and implies exchanges of glances, words, and friction, attractions, and repulsions, demanding us to broaden our perceptual fields beyond immediate home life. It welcomes and hosts us; and it is the necessary second nature that we build to subsist [4] (p. 505). The human being does not dwell, he/she domiciliates, establishes him/herself. The neighborhood is, dual and dynamically, a condition of possibility and a task.

Among neighbors, fraternization occurs, and yet, one remains a stranger, not familiar [5] (pp. 153–154), even unsettling and unpleasant initially, as has been pointed out. However, the step from stranger to enemy is not mandatory [6] (p. 173). It is true that knowledge does not tolerate strangeness; it is hostile to it. But as co-neighbors, in the same way that we are not necessarily friends, we are also not mere visitors. Neighborliness

makes society present and enables hospitality ([2], pp. 35–40). The relational structures of hospitality, while not essential [7] (pp. 223,284), are necessary for our subsistence.

Hospitality says lodging, reception; but it also evokes and implies the hospitable, the therapeutic, and healing—both aspects of some guarantee of affectivity. It is linked with the social value of "phílos" [8] (p. 119) potentially present in neighborliness: it is the promise of reconciliation and it is perceived aesthetically as beauty [9] (p. 34). It makes up hospitality, a positive or enabling aspect of neighborhood, a deflationary cosmopolis [10], a discreet one. One can never resolve the constitutive distance of hospitality between guest and host [11] (p. 227); but in this game, the possibilities we describe arise.

However, being hospitable, we run the risk of not knowing who occupies our home; discouraging potentialities reappear under the shadow of the transgression that inhabits the laws of hospitality. The newcomer who renounces hostility is only a visitor, not yet a guest. The intruder, the meddler, may become what we are by law, but not by nature [12] (p. 63). We should not forget the warning that the other is not the neighbor (πλησίον), but the one who is not one [13] (p. 32).

The relationship among disparities is plural. It could be defined as a 'dualectic' movement. The existence of the otherness only occurs as "being together". Neighborhood is not an organic totality, which articulates its parts instrumentally. It is the articulation itself (of singularities, people, neighbors) which we should retain; not as a mean but as an end. It is the play of the joint itself, of the articulation, that escapes the demands of the totality [14] (pp. 90–91). This common-presence, this appearance, implies a logic that resists both the approach of the dialectical conservation of negativity, which deactivates it, and the mystical enjoyment of the same, which fetishizes it: neither loss nor appropriation, says the mutually exposed condition of singularities [15] that we could name as 'dualectic'. To retain the moment of "externality" is valid essentially, as it no longer refers to any self. The reality is the contradiction itself, mutual belonging, and controversy. The relations of neighborhood consist in the exercise of the right of a particular at the limit of the right of the neighbor, where only mutual respect makes them compatible. This favors the acceptance of difference, avoiding exclusion.

The status of the limit enables intelligibility. The orientation of possibilities related to this limit promotes an ethical uplift; a kind of "frontier ethics" arises [16] (pp. 37–46). "The limit is more than just a sanction; it is where freedom is at stake" ([16], p. 78). The limit is defined by convergence [17] (p. 88) and all commitment is a mutual resignation. Our fellow neighbors or counterparts are not the similar to us, our look-alikes: they are not known but suffered, endured.

Neighborhood is marked by the contiguity of people and the law of touch that governs it, which is also that of separation. To come into contact is not to enter into anything. The being, to exist, needs and seeks to be challenged. It demands otherness or plurality [18] (p. 20). Being is given among us as the event of being in a place, which is a relationship; being is the gift of being located. Showing up, 'comparecencia', is the condition of the singular appearance, and co-presence is the condition of the presence. The logic of co-being is a relation without a relation; a simultaneous exposure to the relationship and the lack of it: the exposure of singularities with one another.

The "with", of co-being, co-presence, etc., is the very structure and tonality of our being. The moment of "with" always transgresses identity, it does not accompany it. A modality of the 'trans' always borders on that of the 'cum'. The co-implication of existing is participation in a world; and a world is nothing external to existence ([15], pp. 45&94). A relationship of correlativity and reversible implication of the unrealized reality occurs. In the ambivalent neighborhood, the other, the contrary or opposite, is not implicated, but complicated [19] (p. 16) and co-implicated: "Things or belongings are constituted in being by implication. Realities are constituted in reality by their co-implication" [20] (p. 155). Our culture cannot develop properly apart from the co-implication of *mythos* and *logos*, image and concept, etc. This co-implication of hope and responsibility within everyday life is where we place the constant origin of the task of reconstructing the human cosmos.

It is the principle that confronts, in human terms, that of non-contradiction, to make it bearable: there is no winner or loser or, properly said, it is winning and losing at the same time. It is the principle of sense, understood as human significance, the dualistic logic [21] (p. 33); it could be the guiding principle of neighborhood, as condition and task—neither monism nor dualism but dualectics. To resolve the antinomy of opposites, what is needed is not a dialectic that overcomes/superates but one that 'suppurates', that is, a classical 'dual-ectic', which is a dialectic that does not explain but co-implicate, capable of reshaping the opposites as *coincidentia oppositorum* (Cusanus), in which the opposites come together but do not merge into something new; there is no synthesis, but coexistence.

It seems to require an analogical relationship between the parts, attentive to the improper (metaphorical) proportionality between the univocal and the equivocal: an open rationality without dispersion; a rigorous one, but without rigidity [22]. The analogy has that non-imposing character, an open and fragile dialectic that aspires to recover the difference by moving away from identity [23]: a dialogical conception that preserves difference without entirely losing identity, through similarity, through analogy. Subtlety is the virtue of hermeneutics [24].

It is in cultural crises, in understanding and communication, where the fusion of hermeneutical horizons thrives. Thinking in terms of "horizons" could be helpful. It is not a rigid boundary but something that invites continuously to enter into it [25] (p. 309). Understanding is the process of merging horizons within the domain of tradition; where the old and the new simultaneously grow toward the truth ([25], pp. 447–458). The horizon of meaning can remain open, and the interpreter's ideas implicated in it. The merging of horizons is a conversation (question–answer). It both limits and enables: "It is more of something in which we make our way and that makes the way with us. The horizon shifts as the one moving progresses" ([25], p. 375). Beyond the politics inherent in this evocative formula, Hans-Georg Gadamer pointed us to Leo Strauss as the political thinker most in line with his hermeneutics. He reminds us that the "you" is not something spoken about but somebody to whom one speaks; and that the relationship between "I", "you", and "we" has a classical name: friendship ([25], p. 633). This assessment goes beyond the purpose of the current article.

### 3. Neighborhood Chores

In this multiplicity of senses, of human significations, we find compassion, not as pity, but as a consequence of the radical contiguity in which we find ourselves ([15], pp. 12 ff.). It is true that in this context, respect for others does not arise so much from fear of internal sanctions or external criticism, but fundamentally from the shame that closeness implies. This also disciplines us [26] (p. 250). The areas of daily life that are not subjected to community scrutiny are very few, although this is more noticeable in premodern or authoritarian societies [27] (p. 281). There are behaviors that are admired, accepted, and despised; these are internalized, not merely on the prospect of hostile reactions [28] (p. 144). This would consist of the internalization of shame; but it is the emotion of self-protection, mutual respect, duty to help, etc., that are more interesting as criteria for action than self-interest, as the latter would vary from one person to another [29] (p. 216)—a kind of "common decency", a pre-institutional and pre-educational moral practice, linked to life in a common space [30].

To lament the evil of the other overlooks the more fearsome evil that its absence would imply; it would mean that the category of the possible itself would crumble. The other is a condition for organization in general. It implies, at the same time, the proposition of a possible world [31] (pp. 353–358). Multiple individual perspectives, in their mutual interplay, make the neighborhood a kaleidoscopic stage. Only this common game of plural and always inexhaustible perspectives, which make up the world as a whole, exist in and for themselves. The world is, primarily for us, that within and amidst which we are; people exist as a horizon [32] (pp. 121–122). Only through the education of everyday life will the other become, at least, an accomplice [33] (p. 21). The otherness would consist in being

together—to be a set and, in this sense, to conform ourselves as singular beings. Against exclusion and/or assimilation, the community claims, as we know, recognition.

Life precedes ideology, as sense precedes reason [34] (p. 191). The neighborhood, in this sense, is prior to the community, although it shares with it its character of gift and infinite task in finitude ([14], p. 47). The community is more than mere gregarious coexistence: it seeks the excess of humanity's never-satisfied lack ([18], pp. 22–23). The neighborhood is not about traditional community, with its potential glorification of land, blood, and identity. This is circumstantial, non-essential; all essential elements are replaceable under certain conditions. Once this identification occurs, the community becomes walled off and separated from its exterior, and the mythical investment is perfectly fulfilled [35] (p. 44). Community and myth speak to each other; this is the most common to its members. The will to the power of myth is doubly totalitarian, in form (will to will) and content (always communion) ([14], pp. 70–71), and can exclude the possibility of community outside of itself. The excess of community is the danger from which politics should protect human life and its interests.

The neighborhood is not the elective community of friendship or lovers, with their potentially subversive anti-sociality. The mandate to love one's neighbor is a discursive excess; it is about loving the well-being of the neighbor, not the person itself [36] (p. 15)—the proximity of the other as pure distance, which is the affirmation of what is distanced itself ([31], p. 208). It is neither the proximity of *gens*, nor *philia*, nor fraternity. Neither community of origin, blood, or principles. It has little to do with the politics of friendship. A friendship must be constantly re-symbolized if we do not want it to run out; the symbol is a testimony of friendship. And these are useful for managing the only possible social bond, that of enmity. Neighborhood is a condition of the possibility of the community. But at the same time, it is delimited by it. Community depends on the redistribution of space, and myth, which is a speech, organizes and distributes the world ([15], pp. 20–61). One could consider the mutual interaction between the neighborhood, material and bodily condition, and the community and its myths. Communities form and deform; they are a living process, but the neighborhood remains, returning to distance and heterogeneity. It acts as a counterpoint to identity whims, returning to the singularity of the concrete. Myth and reasoned discourse are two ways of expressing diverse aspects of human life. Because, without myth, reason-*logos*, becomes instrumental and sterile; without reason-*logos*, myth can turn into a tyrannical illusion. Myths are demythologized to remythologize them according to the new techno-economic and ideological needs of the dominant power.

We are neighbors, perhaps even friends, before becoming citizens. It is interesting to remember how Montesquieu distinguished between laws, which govern the actions of citizens, and customs, which govern the actions of human beings. The neighborhood immediately identifies us after the family nucleus and before school classification. It is the most immediate environment of the individual, where private life is linked with the public one, the co-implication of human space and time, environment and, also, the world of everyday life. It is an existence within the political, but without being determined by legal status. The neighborhood favors a specific community spirit against aggressions. After conflicts, never definitive, these momentary ties are untangled, preventing them from becoming entrenched. Everyday life refers to existence in its very spontaneity. It cannot be captured, it escapes, and it does so because it has no subject: it remains unnoticed. The person on the street is always about to become a political person. This person does not belong to the historical field, even if always ready to burst into history, it remains in the anecdote [37] (pp. 387–392). Everyday life as a place of provisional balance where conflict is latent, implicit, permanently. Everyday life is not just the reproductive function [38] (p. 196), but a potential instrument of change. Groupability is a political construction. And, in everyday life, fear is never alone; it is always accompanied by hope ([35], p. 55). The accomplice, the complicity, is the hope.

While traditional societies presupposed communal structures, modern societies involve collective structures of abstract and atomized individuals. The modern urban fabric

is a framework designed for consumption, not for the relationships [39] (pp. 93–94). On the other hand, the reality is that neighborhoods gradually transform into forgotten suburbs due to the centrifugal dynamics of the city. They do not flee, but are set apart. There is a relationship of estrangement and alienation. The architectural and urban forms of modernity make it difficult to articulate the in-between of the private and the public that politically characterizes the neighborhood. There is a de-structuring of this space. The relationships between neighbors are modified: "The elevator transports its passengers away from the gaze [of others] and deposits them on indistinguishable landings; the similarity between places engenders anonymity" [40] (pp. 106, 107). Even so, the neighbor does not disappear. But he/she hardly coexists and ceases to make the neighborhood a place of mutual knowledge. The neighbor is losing those gratifying words given in the frame of an encounter; those words that give us the feeling of existing itself. With the danger that this implies: where words end, violence begins, which is mute [41] (p. 141).

### 4. Neighborhood as Agora

Against the normalization and homogeneity of consumer society and the mercantile state, the neighborhood, the suburb, is diversity and conflict. In mercantile society there is trade but not community: at most, there is an exchange of good manners and this translates into a lack of sentiment/love ([18], p. 71). Only the experience and negotiation of neighborly conflict, channeled daily, can prevent its sublimation into contained violence and muted aggressiveness. Coexistence in this common, interactive space makes public identity apparent in its dynamic, plural, and impure complexity—also conscious of its irrevocable emotional co-belonging to a place and its projective character. There is a weave, a co-implication, that hides behind each individual and that binds us all together. By speaking and acting, we insert ourselves into a common space where we coexist. Public space makes us responsible for what we say and do and prevents totalitarian whims that lead individuals to hide in private life looking forward, not getting involved in politics. Like it or not, "words watch us" [42] (p. 67).

It is a space where it is easier to distance oneself from the political, not to depoliticize but to resignify it. The neighborhood is also a priority place for politics; it can be conceived of as an agora, understood as an interstitial space between the private and the public. It is a way to avoid the totalitarian tendency to saturate the practical space that power tends to have. It also seeks formulas to articulate the plurality of de facto voices. Neighborhood is a preferred place for political pedagogy for citizenship and involves overcoming the partisan terms of it, focusing on everyday problems. To assume it, that is, to take it into account critically, enables the anchoring of representative democracy to the real ground of human needs, while these enable rethinking it, spurring it on with the daily dynamics of direct democracy derived from the political capillarity of the neighborhood.

The creation of a network of direct democracy mechanisms could be interesting for this. The associative, the familiar, and the personal are examples of non-political domains, but a political conception of them could be and is practicable. Promoting groups of citizen initiatives is not a new idea. Tocqueville already advocated the formation of intermediate structures against the potential apathy of a democracy settled in generalist slogans. Municipal democracy and associativism are fundamental in this objective to avoid administrative despotism. Political participation would thus be mediated by participation in more local organizations, and all interests would be equally represented. Participation would be exercised through a series of intermediate institutions that would interweave personal projects with others that are more common, general, and that would subsequently be incorporated into a common state project. This would be nothing but the result of a debated contract, approved and reviewed by all; otherwise it could become a weapon in the hands of a few people. It is important to remember that the state was born as a community of voluntarily associated property owners. We face a challenging task in this neoconservative landscape.

We could call this neighborly context of dialogue plural speech, since it is not based on the reciprocity of speeches and equality of those who speak, but on a foreign and creative diversity, where it is no longer understood as a "flat geometry" ([37], pp. 144–145); it is an asymmetrical moment, where one does not speak after the other, but we often become confused because we have something in common that does not require regulated language. Conversation with others sustains our world, as no experience is fully real until it has been "spoken" [43] (p. 120). The problem arises when we do not know how to listen, as we expose ourselves to not being heard. Diversity, an essential constituent of human life, is the hallmark of the neighborhood. It is a relatively de-ideologized environment, susceptible to favoring critical awareness alongside debate and the dialogue of shared experiences: a non-community made up of accents, irreducible to a single feature. Being aware of one's own accent helps to embrace strangeness. Listening has a political dimension. Between the world and the human being, language appears symbolically and in reality as a socio-cultural mediation. To philosophically understand the world means "disclosure", the act of clearing and offering clarity. The "being" that can be understood is language.

Neighborhood politics would only make sense by grounding itself in ethics understood as the search for justice, and justice understood as learning to live with the other. It is about promoting coexistence as a first transcendence of the immediate [44] (p. 96)—to build bridges, not to unite but to find (oneself), against politics as a matter of management, administrative, neutral. Living decently is to live with those we have not chosen, whether we like them or not [45] (p. 276).

Individualistic well-being could have serious consequences for citizens by depoliticizing them. Communities are detached from public decisions, as politics have been abandoned to the logic of the private domain. The purpose of politics is the responsibility towards the world. Promoting intermediate structures that structure the social tissue addresses democratic impoverishment. Without active citizenship, institutions run the risk of being emptied of material legitimacy and, without this, legality would be reduced to a mere string of procedural formulas. Institutions degenerate because men and women no longer seek them; uses decline when no one practices them. While formal democracy is formed in vertical consensus from top to bottom, material democracy is based on horizontal consent from bottom to top [46] (p. 95). It is not about agreeing on votes but voting on agreements [47] (p. 96). The choice of democracy is more than a strategy to achieve power, it is a deep political project about the very nature of the society to be built. Following Arendt, it is the great task of learning to live with discrepancies.

## 5. Conclusions

The neighborhood is an ambiguous and ambivalent experience. It has something to do with its co-implicated nature. This is a central category in Andrés Ortiz-Osés' philosophical work and relates to his revision of the dialectical movement called "dualectic"—a more open and antithesis-friendly version of the Hegelian one, following his symbolic convictions: a dialectic without synthesis, open to maintaining tension with alterity, with the opposed opposite, in a creative manner.

As his colleague Eugenio Trías, with whom he shared many subjects and philosophical concerns, expressed in political terms: a deep and productive dialogue with the liminal reality that is the "shadow" (the negative or disharmonious aspects of reality) is necessary. Totalitarian aspirations break this limit, which is "not merely restrictive and negative; nor is it just a barrier to be overcome or an obstacle to be surpassed, transcended, or transgressed. It is, on the contrary, an ontological site of trial (ethical) and definition (of our condition), as well as the determination of a place in a topology that can be thus delineated" [48] (pp. 32,33).

If the neighborhood fulfills or can fulfill the function of a cultural mythology, with its constellation of archetypal symbols and the values derived from them, it should be the next subject to consider regarding the relationship with the philosophy of the Basque-Aragonese author. In either case, although it may be beyond our capacity to determine the

archetypal status of an anthropological experience, we can say that neighborhood would have nothing to envy for the family, so often contemplated by psychology and social and cultural anthropology as the decisive category. Neighborhood entails a plurality of senses, human meanings/significances, which also encourages re-thinking it in symbolic terms.

Considering its characteristics, neighborhood decisively conditions and enables existence while also opening the possibility of influencing it. The relationship with otherness is modulated by it and, without ensuring a cordial encounter, it lays the foundation for a possible dialogue in the interest of a dignified coexistence. Through its symbolic mediation, the boundaries take on a positive character. The neighborhood has enough ontological density to be considered for further development of its implications.

As for the immediate political potentials outlined through the text: in the neighborhood, there is an asymmetric, brief, and discreet dialogue under the unique conditions of proximity/vicinity that define it. Advocating for its characteristics and the mediations that neighborhood life provides, we easily reach refreshing contributions to the democratic system. This is not a nostalgic return to a preconceived and romanticized idea of neighborhood but an attention to its peculiarities and an embrace of them to reconsider our political involvement in common issues, modulate commitments, and reshape higher institutions and practices.

**Funding:** This research received no external funding.

**Institutional Review Board Statement:** Not applicable.

**Informed Consent Statement:** Not applicable.

**Data Availability Statement:** Data is contained within the article.

**Conflicts of Interest:** The author declares no conflict of interest.

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
