# Peer review of "Accomplice Neighborhood: Everyday Life Politics"

_philosophies, doi:10.3390/philosophies8060119_

Round 1

Reviewer 1 Report

Comments and Suggestions for Authors

Peer review:

I believe that this article achieves what it sets out to do: to rethink or restate the theses of the symbolic hermeneutics of A. Ortiz-Osés on the basis of a reflection on the fundamental anthropological experience of neighborhood. Throughout Ortiz-Osés' symbolic hermeneutics we find numerous reflections on politics, although it is not one of his central or fundamental themes. As the author of this article points out, the doctoral theses and monographic studies that have been devoted to the study of this philosopher's work have not directly addressed his interpretation of politics.Therefore, this article is already relevant and important, as it approaches the political, but, in addition, it does so by focusing on the reflection on the neighborhood, which is a topic that has hardly been raised by Ortiz-Osés. In this sense, he makes an original application of symbolic hermeneutics to a new field, enriching it by presenting the neighborhood as a possible site of a new cultural mythology that makes possible and incites to coexist with complicity without the need to agree. In this sense, reflection on the neighborhood from the categories of symbolic hermeneutics could be a way of critically rethinking the democratic system itself.The author of the article shows that he is well acquainted with Ortiz-Osés' philosophy. I consider that he has rightly chosen some of the most characteristic and novel notions of that philosophy, such as "accomplice", "dualectic", "co-implication" to rely on them and elaborate the interpretation of the neighborhood as an intermediate institution between the community and society that allows "building bridges not to join but to meet".Otherwise, the text is very well organized and correctly written, which makes it easy to read and understand. The bibliography is well chosen.

Author Response

Dear reviewer, thank you for your work and guidance. In relation to a political reading of Ortiz-Osés, I believe it would require prior groundwork that I have initiated but is not explicitly detailed here. This brief text can serve as an initial reflection based on this groundwork, which serves as my pretext.
In any case, it is a task I aspire to compose soon, as it is necessary, or at least appropriate (as well as innovative) regarding Ortiz-Osés work.  
I will retain the two aspects you have highlighted (the coexistence in dissent and the issue of encounter without "dissolution" or "communion" of the parties involved) and will endeavor to provide nuances that emphasize them.  
Thank you once again.

Reviewer 2 Report

Comments and Suggestions for Authors

The subject in question on the paper, I have reviewed, could be deepened with references to H.-G. Gadamer, philosopher dedicated in Truth and Method,  to substantiating spaces of interhuman relationships and coexistence with the other.

Author Response

Dear reviewer, thank you for your guidance. I believe that Gadamer's philosophical concept of the "fusion of horizons" would find suitable and relevant placement in the article, along with a potential reference to "Sprachlichkeit" as a condition for the possibility of understanding. Furthermore, considering Gadamer as the indispensable hermeneutic reference for the authors serving me as the pretext for this article, and given his lack of a monographic exploration of political themes (a motivating aspect of my text), I will make a reference to Leo Strauss, whom Gadamer held in high regard in this subject (see "Hermeneutics and Historicism," just before the epilogue of "Truth and Method"). Thank you once again.

Reviewer 3 Report

Comments and Suggestions for Authors

This paper is extremely interesting and has the courage to study a difficult, complex and relatively little known Spanish philosopher at the international level. In this sense, CONGRATULATIONS. 

The main problem I find in the text is the absence of a section in which Andrés' proposal is discussed in detail, making a tour of his research (which is really numerous). Therefore, I suggest that, after the introduction, this section be made and that the most relevant works of the great thinker be specified and put in dialogue with other more internationally known philosophers.

On the other hand, I think that the text lacks a mention to the philosophy of the limit raised by Andrés in, for example, "La razón afectiva". This idea is closely related to the intermediate philosophy of which he speaks in "Cuestiones fronterizas". In this sense, I have found extremely surprising the absence of mention to the analogical hermeneutics of Mauricio Beuchot who in Claves de hermenéutica points out the clear relationship between Andrés' symbolic hermeneutics and analogical hermeneutics. On this aspect I believe that some question should be introduced in the paper. In turn, this philosophy of the limit is related to the work of Lanceros and Trias, in fact this same relation is shown by Andres. Therefore, I think it would be very convenient to delve a little into these issues. It should be taken into account that in that neighborhood mentioned in the article we come up against the limits established by others, which means that we have to opt for a dialogical position in order to establish that "fraterland" (a tentative translation of fratria) of which Andrés speaks. 

Author Response

Dear reviewer,  First and foremost, I would like to sincerely thank you for your kind words and evaluation. Additionally, I will wholeheartedly embrace your proposed translation of "fraterland" as it appears delightful to me.  

A brief introduction placing Ortiz Osés in context is absolutely necessary.  On another note, despite considering myself familiar with his work, I have not delved deeply into "La razón afectiva. Arte, religión y cultura." Nevertheless, I have recently published a brief text on the affective use of reason in Ortiz-Osés, compiling quotes from other works; I will endeavor to incorporate some highlights.  

The explicit recognition of both Beuchot and Trías (especially the latter) in Ortiz-Osés is well-known. Concerning the former, I find the dialectic inherent in analogy particularly interesting, with its metaphorical-metonymic bipolarity, and the author's full awareness that hermeneutics flourishes in crisis (alongside the acknowledgment of its ethical potential). These are the two aspects I will seek to integrate.

Regarding Trías, I will revisit my notes from "La razón fronteriza" to review the aspects related to the limit (which I will project onto the vicinity) and search for any insertable references from "La política y su sombra." Unlike Ortiz-Osés, Trías explicitly articulated his political reflections, and I believe "La política y su sombra" is a relevant book that, although studied, I had overlooked.

Speaking of political thoughts, I believe it is fair to include something related to the greatly admired Vattimo, may he rest in peace. Ortiz-Osés always felt and acknowledged his philosophical and vital affinities with him. I will attempt to make a reference to the "pensiero debole."  

Finally, if I manage to fulfill all the mentioned tasks, I will introduce some idea from professor Lanceros (possibly from "Fuera de la ley"), who inadvertently influenced this article.  

Once again, thank you very much for your work and comments.

Round 2

Reviewer 3 Report

Comments and Suggestions for Authors

This second version has substantially improved the previous paper. Information has been provided on the author studied, which will help readers who do not know him to understand his fratriarchal proposal. In addition, the suggestions indicated have been attended to and the discussion of approaches between the proposal of Ortiz-Osés and that of authors such as Beuchot and Trías has been improved. Undoubtedly we now have before us a work of great interest and clarity.